# A path toward primitive machine intelligence: LMM not LLM is what you need.

## Abstract

We live in a world where machines with large language models (LLMs) and deep reinforcement learning have shown sparks of super-human intelligence in question-answering and playing strategic boardgames. At the same time, animals continue to reign supreme in the sense of smell, a primitive form of intelligence. Applying the former (deep learning) tricks on large datasets of hyperspectral hardware and spectrometers may well lead to artificial noses that can detect the chemical composition of a mixture. But it comes at the cost of interpretability!

Here, I propose a path that uses linear mixture models (LMMs) to build an engineering theory of cognitive development for chemosensing. With creative mathematical models, we can derive analytical expressions for the limits of chemosensing and advance the statistical mechanics of learning.

## 1 Introduction

Look at Patron (in Figure 1), a dog that is energy-efficient, portable, and parts-per-trillion sensitive to detect underground explosives by their chemical traces in the air. Machines such as spectrometers used in analytical chemistry (check Table-1 of Sharma et al. (2023)), are no match for sniffer dogs yet. In gas-phase spectrometry, the measured spectrum of a mixture of chemicals is often a linearly weighted sum of a basis of chemical spectra. Similarly, for imaging spectrometers found in microscopes and remote sensing satellites, the measurement at a pixel of the hyperspectral camera is a linear combination of the spectra of materials located inside the area covered by the pixel. Now, despite the mathematical simplicity of a linear mixture model (LMM), learning to unmix spectral signals is challenging in practice due to the presence of sensor noise, photon noise, low concentrations of target chemicals, and interference from a large number of background chemicals (Duarte et al., 2014). Here, we shall build a cognitive theory for chemosensing to calculate some minimum hardware requirements for dog-level performance. One day, this could lead to the development of artificial noses with super-dog intelligence, learning to detect chemicals much faster than animals and in a more interpretable way than chemometric classification using deep learning (Deshpande et al., 2021; Hong et al., 2021a;b).

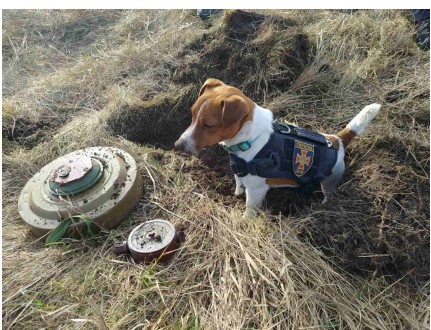

Figure 1: Patron, dog mascot of the State Emergency Service of Ukraine. (Wikimedia Commons)

Table 1: Stages of chemosensing

| | |
|---|---|
| Sensory | Infer the number of chemicals in a set of mixtures |
| Preoperational | Learn spectral representations of the chemicals |
| Concrete operational | Estimate the concentration of a chemical in a mixture |
| Formal operational | Interpret why a mixture is good or bad by reasoning from past examples |

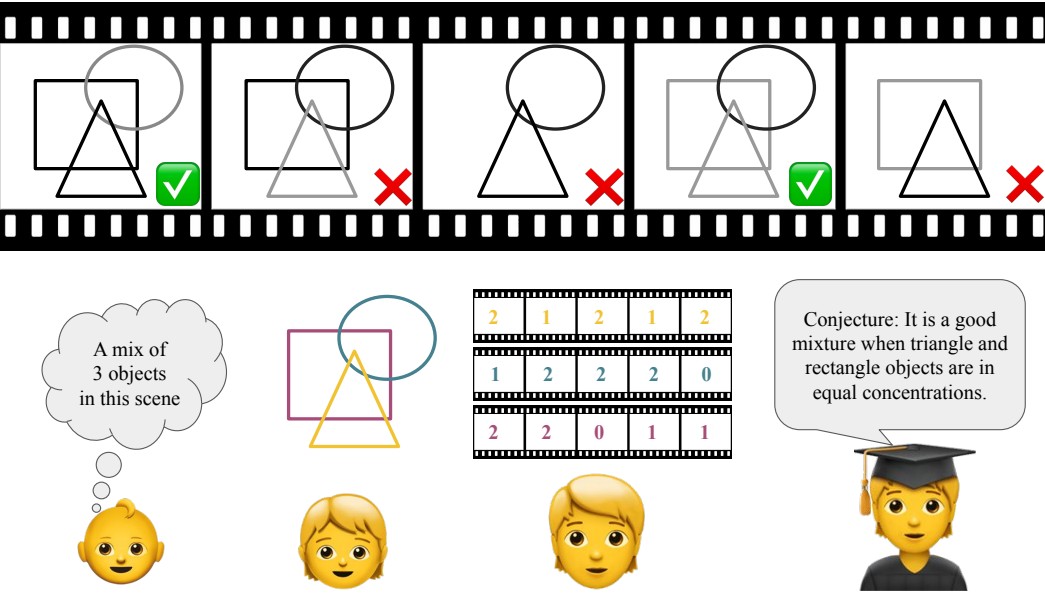

Figure 2: The four stages of cognitive chemosensing. Chemical spectra are depicted abstractly by objects of different shapes, so that a mixture of chemicals of varying concentration is a frame of overlapping objects with varying intensity.

## 2 COGNITIVE THEORY OF CHEMOSENSING

Let us take inspiration from Jean Piaget's theory of cognitive development (Piaget & Cook, 1952), which Einstein is quoted to have called a discovery "so simple that only a genius could have thought of it". Piaget experimented on children and formulated that the development of reasoning occurs over four stages, a theory that is textbook knowledge for psychologists but **lacks a mathematical demonstration**. Here, I adapt his **stage theory to an abstract world of chemical mixtures** (see Table 1 and Figure 2). As a thought experiment, imagine a collection of Raman spectra measured on plant leaves (Gupta et al., 2020). Now, there are likely more than 1000s of different kinds of proteins, carbohydrates, fats, and micronutrients in that leaf whose pure spectra are **linearly mixed** to result in the measured spectra. Not all chemicals can be objectified, especially those in ultra-low concentrations or those with pure spectra similar to the pure spectra of high concentration chemicals. Thus, the first stage is to infer the number of objectifiable chemicals in the collection of spectra. The second stage is to infer a symbolic representation, i.e. the pure spectrum of each chemical whose presence was inferred. The third stage is to quantitatively estimate the concentration of each known chemical in the mixed spectra. The fourth stage is to scientifically explain the nutritional goodness of a leaf given labelled examples. This kind of a problem formulation has been long awaited, as Svante Wold (who coined the term chemometrics) said over 30 years ago (Geladi & Esbensen, 1990): "In this century, the most important thing we have learned is design. How to set up an experiment. And in the next century, we may learn how to formulate a problem in a meaningful way. If you consider that as a part of chemometrics, then it is the most important. But I think it resides outside chemometrics. Problem formulation is important in all branches of science". With this motivation, we shall pioneer a path where LMM is what (if not, all) you need.

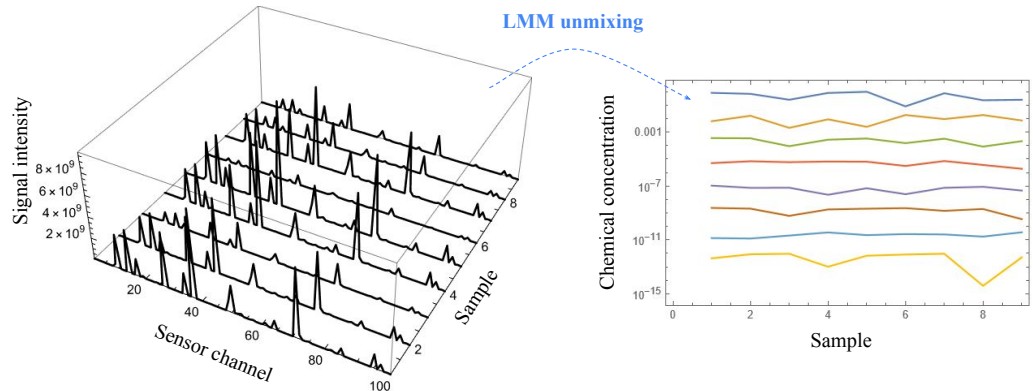

Figure 3: Linearly mixed observations and their underlying chemical concentrations. All simulations are done using Wolfram Language 13.1.

In our path, we shall commit to foundational and analytical theory instead of hunting for narrow-scope empirical successes on real-world datasets. Hence, I propose that we build and validate theories using synthetic datasets based on Algorithm 1 with the hardware parameters $(M, N, R, S, T)$ defined in the comments, the pure spectrum $(\vec{a}_m)$ of a chemical $(m)$ mathematically modelled (based on intuition gained from observing signals at spectrabase.com) to be a random permutation of a list of exponentially increasing numbers, and the chemical concentrations $(p_m(t)$ for $m = 1 : M)$ sampled from uniform distributions of exponentially decreasing widths (because we focus on a parts-per-trillion sensitivity). In Figure 3, we plot the calculated linearly mixed observations $(\vec{s}(t)$ for $t = 1 : T)$ and the underlying chemical concentrations. This is a dataset for LMM unmixing that is non-trivial, and yet small enough to embrace researchers with only access to low-compute resources. The dataset and Wolfram Language code for our subsequent analysis, is provided in the **supplementary file**. This would be the **first ever dataset for linear unmixing** at https://paperswithcode.com/datasets. It would also be of interest to the interpretable machine learning and unsupervised few-shot learning communities, which in my humble opinion, are often fighting a pre-mature battle against highly nonlinear datasets such as image-net.org.

---

**Algorithm 1** Simulate LMM given hardware parameters

---

1: **Default parameters**
2: $M = 8$      ▷ number of chemicals in the Mixture
3: $N = 100$      ▷ Number of sensor or spectral channels
4: $R = 10^{10}$      ▷ dynamic Range of a pure spectrum
5: $S = 10^{-12}$      ▷ Sensitivity of chemical concentration standard deviations
6: $T = 9$      ▷ number of samples over Time
7: **procedure** SYNTHETICLMM($M, N, R, S, T$)
8:      $\alpha = R^{1/(N-1)}$
9:      $\beta = S^{1/(M-1)}$
10:      **for** $m = 1$ to $M$ **do**
11:          $\vec{\pi}_m = $ RandomPermutation($0$ to $N - 1$)
12:          $\vec{a}_m = \alpha^{\vec{\pi}_m}$      ▷ Generate pure spectrum
13:      **end for**
14:      **for** $t = 1$ to $T$ **do**
15:          **for** $m = 1$ to $M$ **do**
16:              $p_m(t) = $ UniformRandomNumber($0, \beta^{m-1}$)      ▷ Generate chemical concentration
17:          **end for**
18:          $\vec{s}(t) = \Sigma_{m=1}^{M} p_m(t)\vec{a}_m$      ▷ Calculate signal intensity
19:      **end for**
20: **end procedure**

---

Let us now begin with the analysis of our model and dataset over the four stages.

## 2.1 Infer the number of chemicals in a set of mixtures

Inferring the chemical dimension $M$, just from the observations in Figure 3 is a classic problem with a long history (Malinowski, 1977) of performing an eigenvalue cutoff of the sample covariance matrix $\mathcal{Q}$. However, this has to be done carefully. In particular, the observations have to be rescaled to have a unit mean value at each sensory channel and large values for $N$ and $R$ are necessary for the correct estimation of $M = 8$, as depicted in Figure 4. An analytic justification is sketched below.

**Proof of eigenvalue cutoff in rescaled sample covariance matrix:** $\mathcal{Q}(\ \vec{s}(t = 1 : T)/\ \vec{\bar{s}}\ )$ has the elements $q_{jk} = \Sigma_t (s_j(t)/\bar{s}_j - 1)(s_k(t)/\bar{s}_k - 1)/(T-1) \overset{T \gg 1}{\approx} (\overline{s_j s_k} - \bar{s}_j \bar{s}_k)/(\bar{s}_j \bar{s}_k)$ where $\bar{x}$ denotes the mean value of $x(t)$. In our model, $\bar{s}_n = \Sigma_m \bar{p}_m a_{nm} = \Sigma_m \beta^{m-1} a_{nm}/2$ and $\overline{s_j s_k} - \bar{s}_j \bar{s}_k = \Sigma_m \Sigma_{m'}(\overline{p_m p_{m'}} - \bar{p}_m \bar{p}_{m'}) a_{jm} a_{km'} = \Sigma_m (\overline{p_m^2} - \bar{p}_m^2) a_{jm} a_{km} = \Sigma_m \beta^{2m-2} a_{jm} a_{km}/12$. For $\alpha \to \infty$, we can show that $q_{jk} \approx 1/3$ only for $j = k = n$ with $a_{nm} = R$, otherwise $q_{jk} \approx 0$. For $N \gg M$, it is thus almost certain that there are exactly $M$ non-zero elements in $\mathcal{Q}$, which are all in the diagonal.

Future work to evaluate methods for dimensionality estimation may also include noise added to the signals produced by Algorithm 1. We may begin with old methods such as the Malinowski indicator function, Akaike information criterion and minimum description length (Chang & Du, 2004); then Hyperspectral signal identification by minimum error (HySime) (Bioucas-Dias & Nascimento, 2008), which is the only method to correctly infer the dimensionality of the Indian Pines hyperspectral image segmentation dataset; and random matrix theory (Cawse-Nicholson et al., 2013) which works better or worse than HySime depending on the nature of the correlated noise in the dataset.

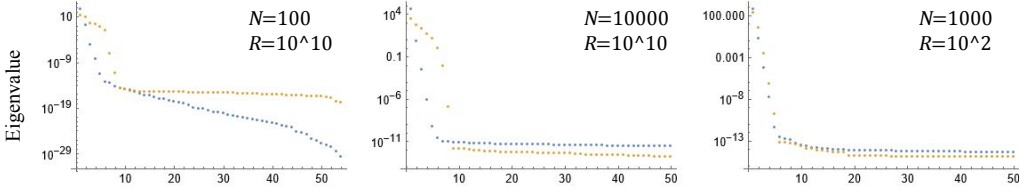

Figure 4: Eigenvalue decay plot of the sample covariance matrix of the observations (blue) and unit-mean rescaled observations (yellow). In addition to the observations shown in Figure 3, two other observations under different hardware constraints are considered.

## 2.2 Learn spectral representations of the chemicals

For the LMM, since there are $MN + MT$ variables and $NT$ equations, we get an overdetermined system for $T > MN/(N - M)$ (and thus for $T = M + 1$ if $N > M(M + 1)$). In principle, this system can be solved consistently to get an accurate estimation of $\vec{a}_m$. However, the Wolfram function repository's default methods for nonnegative matrix factorization (NMF) (Lee & Seung, 1999) and independent component analysis (ICA) (Herault & Jutten, 1986) did not converge to the pure spectra for the observations in Figure 3 (using $M = 8$) and perform worse than principal component analyis (PCA) based on the eigenvalues of the sample covariance matrix (see results in Figure 5). This opens up a **concrete playground to refine our mathematical understanding of ICA and NMF**. Analytic convergence laws may be reverse engineered with the intuition gained from simulations, and the computational efficiencies over alternatives such as singular value decomposition, pixel purity index, and maximum distance methods (Bajorski et al., 2004) need to be explored.

Note that, ICA can also be implemented as a neuromorphic signal processing method to directly learn the ideal pseudo-inverse $\vec{w}_{1:M}$. Ideally, $\vec{w}_m \cdot \vec{a}_m = 1$ and the vector-sum $\Sigma(\vec{w}_m) = 0$ to cancel out the noise. Thus, when $N \gg M$, we can expect that $\vec{w}_m \approx \frac{\vec{a}_m - \Sigma(\vec{a}_m)/N}{\vec{a}_m \cdot \vec{a}_m - \Sigma(\vec{a}_m)^2/N}$. This can be further simplified when $N \gg \ln R$, because $\Sigma(\vec{a}_m) = \Sigma_n \alpha^{n-1} = \frac{R^{N/(N-1)} - 1}{R^{1/(N-1)} - 1} \approx RN/\ln R$ and $\vec{a}_m \cdot \vec{a}_m = \Sigma_n \alpha^{2(n-1)} \approx R^2 N/\ln R^2$.

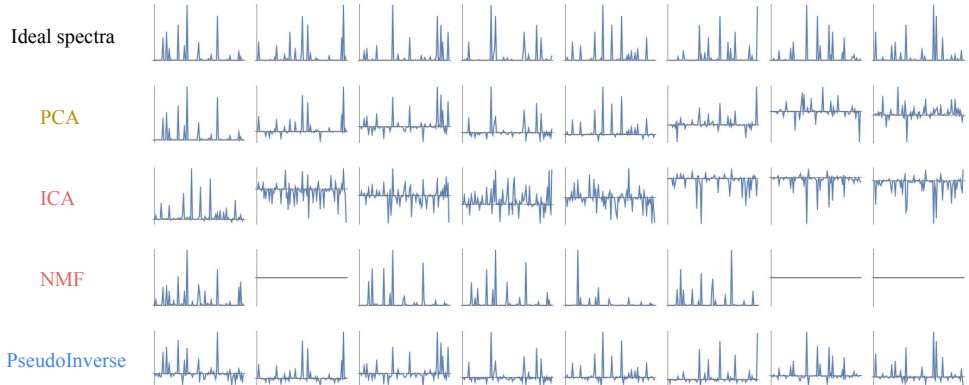

Figure 5: None of the existing methods are able to learn the pure (ideal) spectra, despite the existence of its pseudo-inverse. Novel update rules for ICA and NMF are needed to solve this problem.

### 2.3 ESTIMATE THE CONCENTRATION OF A CHEMICAL IN A MIXTURE

A dot product of the observations (Figure 3 left) with an ideal pseudo-inverse spectrum (Figure 5 bottom) will give an estimate of the underlying chemical concentration (Figure 3 right). The maximum absolute error in estimation due to floating-point arithmetic is found to be $\Delta_{\max} \approx 10^{-15}$, thus guaranteeing parts-per-trillion sensitivity.

**Error scaling law**: $\Delta_{\max} < N \times \text{Precision} \times \max s \times \max w$.

Using the simplification from Section 2.2 for $\vec{w}_m$ when $N \gg M$ and $N \gg \ln R$, we can obtain $\max w \approx g(R)\frac{\ln R}{RN}$ where $g(R) \equiv \max(\frac{1-1/\ln R}{1/2-1/\ln R}, \frac{1-R/\ln R}{R/2-R/\ln R})$. Points of interest are that $g(R \gg 1) \approx 2$, $g(R = e^2) \gg 1$, $\min g(R) \approx 0.9$ at $R \approx 1.4$, and $g(R = 1 + \epsilon) = 1$. For $M \gg 1$, we obtain $\max s \approx R + \Sigma_{m=2}^M \beta^{m-1}\mathbb{E}a \approx R(1 + \frac{M}{\ln 1/S \ln R})$; and for $M \ll \ln R \ln 1/S$ we have $\max s \approx R$.

In our case, $N = 100$, $\text{Precision} = 1.11 \times 10^{-16}$ (for 64-bit float), $\max s \approx 10^{10}$ and $\max w \approx 2\ln 10^{10}/10^{10}/100 \approx 4.6 \times 10^{-11}$ yields $\Delta_{\max} < 5.1 \times 10^{-15}$. This clears the way for future work which may include an investigation of the limits of accurate estimation in the presence of noisy observations. Note that matched subspace detectors (Scharf & Friedlander, 1994) are understood with Gaussian noise, but spectral signals are often dominated by Poisson noise.

### 2.4 INTERPRET WHY A MIXTURE IS GOOD OR BAD BY REASONING FROM PAST EXAMPLES

Ideally, LMM unmixing results in the unsupervised learning of interpretable features that is out-of-distribution generalizable (Ye et al., 2021). To establish this, we need to test convenient mathematical models to develop intuitions and use real-world open-access datasets to provide proof-of-concepts. An empirical demo may even require the development of new hardware technology to extend the linear dynamic range of its sensors.

Note that this final stage of learning can be compared to learning in the association-response layers of perceptrons (Rosenblatt, 1961). While the sensory-association layer of classical perceptrons was untrained and randomly connected; here, the methods of Section 2.1 fix the number of association neurons and the methods of Section 2.2 fix the weights of the sensory-association layer. Also the association neurons are linear, and the nonlinearity is only in the association-response layers. I recommend that we focus on datasets where the response units are binary, for example using spectral measurements on the skin to detect skin diseases. But, of course this approach can also be extended for non-binary datasets such as predicting pH levels of farmland patches using hyperspectral images (https://zindi.africa/competitions/geoai-challenge-estimating-soil-parameters-from-hyperspectral-images), where interpretable learning may require much larger datasets to sample across different chemical species that could result in the same pH.

## 3 Conclusion

We have now perhaps the **first mathematically tractable framework for cognitive development**, by a dogged focus on chemical mixture understanding, a primitive form of intelligence. Hope this is complementary to other paths (LeCun, 2022) that focus on more advanced and autonomous intelligence such as natural language understanding and motor development. Note that, apart from spectrometers, LMM theory is also applicable for the analysis of gene-expression microarrays (Bazot et al., 2013) and gas-sensing material arrays (Persaud & Dodd, 1982). A friendly warning to new entrants of the LMM world, it should not be confused with a linear mixed-effects model (Lippert et al., 2011; Gałecki et al., 2013), which is popular in bioinformatics.

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
