# OpenReview forum: "A path toward primitive machine intelligence: LMM not LLM is what you need."
_ICLR.cc/2024/Conference — Submitted to ICLR 2024_

### Official Review · Reviewer_wJx4 · 2023-10-23

**Soundness:** 2 fair
**Presentation:** 2 fair
**Contribution:** 1 poor
**Rating:** 1
**Confidence:** 5

**Summary:**

This paper outlines some perspectives on the task of chemosensing, where one identifies the chemicals present given sensory data — artificial smelling of sorts. The paper (1) briefly describes a four-stage cognitive theory of chemosensing based on Jean Piaget's theory of cognitive development, (2) describes classical techniques for inferring the number of chemicals in a set of mixtures based on using eigenvalue cutoffs, PCA, or nonnegative matrix factorization, and (3) briefly discusses other miscellaneous aspects of the problem.

**Strengths:**

The prose is clear, easy to follow, and enjoyable to read. The authors provide helpful figures, use sections and subsections well.

**Weaknesses:**

The paper is thin on material. There is a very short explanation of related work, and appears to be no "contributed work" in the paper: no dataset description (although a working example is briefly described), no techniques proposed, no evaluations, etc. It is a brief discussion of the problem of chemosensing.

I think the area of chemosensing could make for a good machine learning paper, but doing so would require:
- A longer section on related work, contextualizing your approach
- Creating or using a dataset for this task with clear evaluation metrics
- Defining an approach, implementing it, and evaluating it on your dataset
- Show experiments comparing to baselines and understanding how your approach works

**Questions:**

None

---

> ### Author Response · Authors · 2023-11-22
>
> Thank you, I believe I have addressed all the weaknesses you pointed out. I hope this paper can be accepted and its dataset steer the community towards the challenging task of linear unmixing.

---

### Official Review · Reviewer_gSc5 · 2023-10-24

**Soundness:** 1 poor
**Presentation:** 2 fair
**Contribution:** 1 poor
**Rating:** 1
**Confidence:** 3

**Summary:**

This paper proposes a path towards primitive machine intelligence using linear mixture models (LMMs) for chemosensing. The authors argue that while machines with large language models and deep reinforcement learning have shown sparks of super-human intelligence in certain tasks, animals still reign supreme in primitive forms of intelligence such as smell. By using LMMs, the authors aim to build an engineering theory of cognitive development for chemosensing and advance the statistical mechanics of learning. The paper contributes to the development of a mathematically tractable framework for cognitive development and provides a complementary approach to other paths that focus on more advanced and autonomous intelligence such as natural language understanding and motor development.

**Strengths:**

**Originality**
Olfaction is an understudied problem, particularly at a venue like ICLR, and any steps towards better understanding it from a machine-learning perspective would likely be novel.

**Quality**
The experiments that are conducted appear sound with appropriate parameter choices.

**Clarity**
Overall, the paper is written in an engaging way. Most statements in the paper are generally clear. Figure 2 is a nice conceptual diagram.

**Signifiance**
As mentioned above, machine olfaction is relatively understudied. Any steps toward a theoretical understanding of it would be significant to the field.

**Weaknesses:**

In my view, unfortunately, there are several issues with this submission:

First, coverage of related work is not adequate. It is unclear what the prior models of chemosensing are, and how the proposed model differs from prior work.

Secondly, the content of the contributions seems to be lacking. Frankly, beyond motivating the problem that existing methods are unable to learn the true spectrum, I'm struggling to see what this submission contributes, although I may be missing some key aspects of the submission. One possibility to improve this submission's contribution is to derive an analytical scaling law as suggested in Section 2.3.

Third, it is mentioned that one advantage of LMMs is their interpretability. It would be ideal to investigate the interpretability of LMMs experimentally.

Also, the notation is a bit unclear. I would suggest defining the dimensionalities of all the variables as well as making sure all the variables are defined as soon as they are introduced. The author may wish to consider adding a notation table.

**Minor Comments**

"worser than" in Section 2.2

The parameters in Table 2 are not justified or explained

**Questions:**

How does this work relate to prior work? What is the state of our computational understanding of chemosensing? What is the prior theory on LMMs?

What are the key contributions of this paper?

---

> ### Author Response · Authors · 2023-11-22
>
> Thank you, also for your minor comments. I hope you enjoy the analytic results I have now provided (as you requested) and shall revise your rating for the upgraded paper.

---

### Official Review · Reviewer_G4jL · 2023-10-26

**Soundness:** 1 poor
**Presentation:** 1 poor
**Contribution:** 1 poor
**Rating:** 1
**Confidence:** 4

**Summary:**

Not applicable

**Strengths:**

not applicable

**Weaknesses:**

inacceptable as is

**Questions:**

no questions - just an inappropriate paper

**Details Of Ethics Concerns:**

this paper is apparently a joke.
The paper writes about a "dogged focus on chemical mixture understanding".
It randomly combined text passages.
Details are inappropirate and incomprehensibly combined...

---

### Official Review · Reviewer_j3FH · 2023-12-05

**Soundness:** 1 poor
**Presentation:** 1 poor
**Contribution:** 1 poor
**Rating:** 1
**Confidence:** 4

**Summary:**

This paper seems somewhat AI-generated, and at the very least not a serious paper. The paper proposes chemo-sensing algorithms but does not provide a concrete problem statement, method, or evaluations beyond some semi-coherent plots. The references are not apt. Overall, I am absolutely not certain this paper passes the Turing test, and definitely not the bar for acceptance at ICLR.

**Strengths:**

N/A

**Weaknesses:**

See the summary section.

**Questions:**

* What is the key contribution in this work?
* What are the baselines?
* How does the paper improve upon them?

**Details Of Ethics Concerns:**

Seems like a joke/AI generated submission, and I would urge the committee to consider the ethical implications of submitting such a work.

---

> ### Public Comment · ~Celestine_Preetham_Lawrence1 · 2024-05-14
> **reposting the comment from 05 Dec 2023 (no idea why it is not public unlike other replies to reviewers)**
>
> Which reference is not apt? Just because I write in an original way, does not mean the paper is AI generated - I do not even use tools like Grammarly on my papers. Can you be objective instead of throwing insults? The ethics chair should definitely look into your kind of subjective reviews, especially when the paper has strong mathematical results that you have completely overlooked.
>
> What is the key contribution in this work?
>
> * Supplementary file has the first ever dataset for linear unmixing at https://paperswithcode.com/datasets if accepted
> * I claim this is the first mathematically tractable framework for cognitive development because of the following sub-contributions -> Figure 2. introduces an adaption of Piaget's stage theory of cognitive development for chemical mixture understanding -> Section 2.1 contains a mathematical proof for the empirical finding (plotted in Figure 4) of eigenvalue cutoff being more pronounced for the rescaled sample covariance matrix -> Section 2.2 shows that ICA and NMF fail for the simple linear unmixing dataset (and I am willing to test it again any other state-of-the-art algorithm proposed and accepted in ICLR2024) -> Section 2.3 now contains an error-scaling law as requested by Reviewer gSc5
>
> What are the baselines? In Section 2.2 the ideal result is the PseudoInverse solution. The baseline method is PCA. ICA and NMF needs new theory development to beat the baseline, and attain the ideal result. Thus, I expect the dataset provided in this paper to be a simple yet profoundly impactful challenge for the learning theory community.
>
> How does the paper improve upon them? This paper is the first to define the problem. My bet is that many of the algorithms proposed in ICLR 2024 for variations of NMF would still fail against this dataset. Hence, it is important that this paper is accepted so learning theorists, even with low-compute resources, have a principled dataset to play with and innovate.

---

### Author Response · Authors · 2023-11-22

Thanks to Reviewer-gSc5 and Reviewer-wJx4. I have updated the manuscript based on your detailed feedback and have now made my contributions clearer as bold text. I hope that you will be proud of the foundational mathematical contributions of this paper. I am willing to compare it with other learning-theory related ICLR 2024 papers in review, if accepted. Your suggestions are welcome!

https://openreview.net/forum?id=z8q8kBxC5H
https://openreview.net/forum?id=lNCnZwcH5Z
https://openreview.net/forum?id=DchC116F4H
https://openreview.net/forum?id=fzc3eleTxX
https://openreview.net/forum?id=AcGUW5655J
https://openreview.net/forum?id=8JKZZxJAZ3

---

### Public Comment · ~Celestine_Preetham_Lawrence1 · 2024-05-14
**Peer review in a globalized world**

The conference organizers should be ashamed that a Meta Reviewer uses such unjustified language. I have now decided to abandon theoretical research and develop software for a private company.

---
P.S. I am not the only one lamenting about the review process, it can never be perfect but I would like to stress again the following words from a senior researcher (https://openreview.net/forum?id=DchC116F4H&noteId=7sF5dWABhU): "A good review process is one that serves two overlapping functions. From the conference perspective it should identify the most relevant/excellent/solid/important manuscript for participation in the meeting. From the author’s perspective it is a chance to get unfiltered but hopefully constructive critique that will allow us to improve our science. In my view these two functions are tied together, in that a good constructive review is one that shows the program chairs what are the merits and shortcomings of the paper and allow them to balance these in the bigger view of other submissions. Except for extreme cases, a terse review that does not provide information is also one that is not useful for the purposes of decision making."

P.P.S. I would recommend that ICLR2025 implement LLM technologies to filter out poor comments. I give below an example using Bing AI.

*Query*: Is the following review acceptable for a scientific conference : "This paper seems somewhat AI-generated, and at the very least not a serious paper. The paper proposes chemo-sensing algorithms but does not provide a concrete problem statement, method, or evaluations beyond some semi-coherent plots. The references are not apt. Overall, I am absolutely not certain this paper passes the Turing test, and definitely not the bar for acceptance at ICLR." ?

*Answer*: The review provided does offer some critique of the paper but it could be improved to be more constructive and specific. Here are some points to consider:

**Specificity**: The reviewer should provide specific examples from the paper to support their claims. For instance, they could point out which parts of the paper seem AI-generated or which references are not apt.

**Constructiveness**: The review should aim to help authors improve their work. Instead of just stating that the paper does not provide a concrete problem statement or evaluations they expected, the reviewer could suggest what kind of problem statement or evaluations they expected to see.

**Professionalism**: The tone of the review should be professional and respectful. Phrases like “not serious” or “I am absolutely certain this passes the Turing test” could be perceived as dismissive or disrespectful. It’s important to critique the work, not the authors.

**Actionable Feedback**: The reviewer should provide actionable feedback that authors can use to improve their papers. This might include suggesting additional experiments, recommending relevant literature for citation, or offering ideas for clarifying the presentation of material.

Remember, the goal of a review is not just to judge the quality but also to provide feedback that helps authors improve their work. Good reviews are specific, constructive, professional, and provide actionable feedback.

---

### Meta-Review · Area_Chair_q2oP · 2023-12-05

**Metareview:**

Reviewers are in consensus that the submission does not make a clear contribution and that the engagement from the author(s) is disingenuous. The conference should consider adopting a policy for spam submissions next year.

**Justification For Why Not Higher Score:**

spam submission

**Justification For Why Not Lower Score:**

N/A

---

> ### Public Comment · ~Celestine_Preetham_Lawrence1 · 2024-05-14
>
> Calling work that stems from years of insight as a "spam submission" is just killing any future motivation to pursue a research career. This is an unacceptable culture.

---

### Decision · Program_Chairs · 2024-01-16

Reject